# Copywriters' preference evaluation on online copywriting course attributes during the COVID-19 pandemic

Cheselle Jan Roldan[1,2], Yogi Tri Prasetyo[1,3,4]*, Ardvin Kester S. Ong[1], Irene Dyah Ayuwati[5], Satria Fadil Persada[6], Reny Nadlifatin[7]

1 School of Industrial Engineering and Engineering Management, Mapúa University, Intramuros, Manila, Philippines, 2 School of Graduate Studies, Mapúa University, Intramuros, Manila, Philippines, 3 International Bachelor Program in Engineering, Yuan Ze University, Chung-Li, Taiwan, 4 Department of Industrial Engineering and Management, Yuan Ze University, Chung-Li, Taiwan, 5 Department of Information System, Institut Teknologi Telkom Surabaya, Surabaya, Indonesia, 6 Entrepreneurship Department, BINUS Business School Undergraduate Program, Bina Nusantara University, Jakarta, Indonesia, 7 Department of Information Systems, Institut Teknologi Sepuluh Nopember, Surabaya, Indonesia

* yogi.tri.prasetyo@saturn.yzu.edu.tw

**Data Availability Statement:** The data of this study is available online through this link: 10.6084/m9.figshare.22577749.

## Abstract

Copywriting online course has become a famous online training over the past years and the reliance on online courses increased even during the COVID-19 pandemic. In recent years, online courses have become a popular training platform, especially for copywriting courses. The demand for online courses increased during the COVID-19 pandemic, prompting the need to optimize the learning experience of an online course's target audience. This study aimed to determine the combination of online course attributes most preferred by Filipino copywriters such as course style, payment method, course delivery, module duration, and course type. 292 Filipino copywriters from a leading Philippine-based copywriting group voluntarily participated in this study and answered an online questionnaire quantitative survey which was distributed using the purposive sampling method. Conjoint Analysis with an orthogonal design revealed that copywriters consider the course style attribute as the most important (46.007%), followed by payment method (18.236%), and course delivery (15.435%). Module duration (10.489%) and while the course type (9.833%) were was the least considered attribute of an online course. The result shows that Filipino copywriters prefer an intermediate-level video course on a Facebook group that lasts 1 to 3 hours per module and is paid per course for a total utility score of 0.281, while the least preferred combination was a beginner-level audiobook course that lasts less than 30 minutes per module, delivered via email, and paid per module, for a total utility score of -0.281. This study is the first study that analyzed the copywriters' preference for online copywriting course attributes during the COVID-19 pandemic. The findings of this study are beneficial to online course creators who are targeting copywriters. Finally, the result of this study can be expanded further to other online courses worldwide.

**Funding:** this research was funded by Mapúa University Directed Research for Innovation and Value Enhancement (DRIVE). The funders had no role in study design, data collection and analysis, decision to publish, or preparation of the manuscript.

**Competing interests:** The authors have declared that no competing interests exist.

## 1. Introduction

Copywriting is the practice of using the written word to persuade readers into taking action, usually for sales, advertising, and marketing purposes [1]. People who are employed or are earning a living by specializing in this type of writing skill are called copywriters [1]. With the advent of the internet and social media, the span of applications of copywriting expanded beyond slogans, commercials, newspaper ads, and billboard ads to social media ads, product descriptions, emails, and web pages [2]. "Indeed" being one of the leading global employment websites determined 3 main types of copywriters: agency copywriters–who write for multiple brands and clients handled by their agency, corporate copywriters–who only write copy for their employer's products, and freelance copywriters–who offer their copywriting services on the market independently [3]. In recent years, the 3 main types of copywriters are dominated by freelance copywriters due to the emergence of freelancing platforms such as Upwork and Freelancer.

Today, copywriting is ranked 3rd among the top 10 high-paying freelancing skills, especially during the COVID-19 pandemic [4]. Moreover, copywriting is projected by the U.S. Chamber of Commerce as one of the most in-demand service-based businesses in 2020 [5]. These are all understandable rankings and projections as businesses all over the globe are being forced by the COVID-19 pandemic to finally consider adopting a digital approach. From a previous digital adoption rate of 35% in 2019, the global adoption rate spiked up to 55% [6], with the Asian-Pacific Region making the most significant leap from 33% to 54% (Fig 1).

The same pandemic that led to the surge in digital adoption rate across the globe also led to 4.5 million Filipinos facing the unemployment line, the most the Philippines has suffered in the past decade and a half [7]. The unprecedented effect of the COVID-19 pandemic on the Philippines also marked its first economic contraction in 20 years [7]. These unfortunate situations forced laid-off Filipino workers and Filipinos worrying over their job's security to consider online freelancing in a bid for survival. To underscore this massive transition, "Payoneer", an online money transfer and digital payment service provider, reported that the Philippines showed a 208% online freelancing growth [8] in 2020, which is indicative of ample supply to meet the demand generated by the growing digital adoption rates of businesses globally.

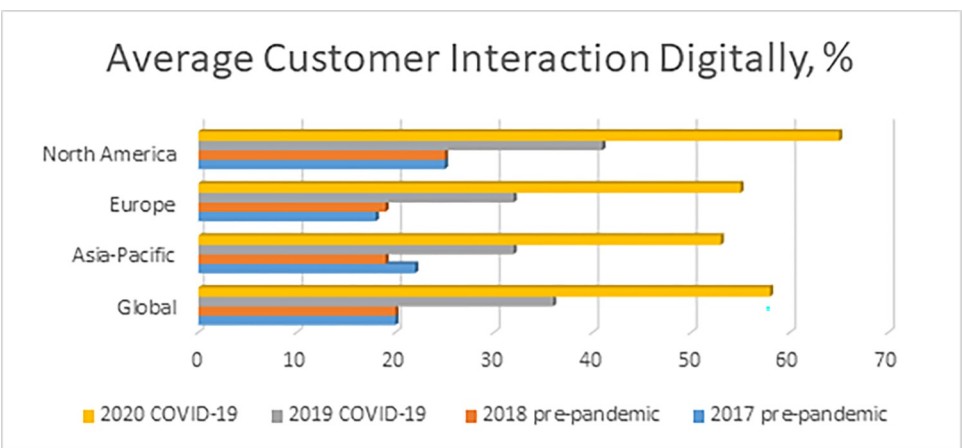

**Fig 1. Significant increase of digital adoption of businesses globally during the COVID-19 crisis** [6].

Some of the most in-demand skills for Filipino freelancers are programming, graphic design, SEO, and English writing [9]. For Filipinos considering freelancing as an additional income source, the skill they can feasibly offer to start freelancing as soon as possible is English Writing as the skill entry barrier is objectively lower, considering that the Philippines's English Proficiency Index in 2020 is 562, the second-best in all of Asia and 27th in 100 countries [10]. This makes copywriting a more lucrative option for Filipino freelancers who are looking to generate income from their English writing skills. However, learning copywriting is primarily accessible only to Filipinos earning their bachelor's degree in advertising. To overcome this curriculum hurdle and the present learning strain caused by the inevitable elimination of face-to-face classes brought by the COVID-19 pandemic, the best copywriting learning option for aspiring and practicing Filipino freelancers is online learning.

Udemy, one of the most popular online course platforms with millions of students enrolled, reported a significant growth in enrollment of technical skills geared mainly towards AI and machine learning, and a significant growth in soft skills such as growth mindset, creativity, focus, innovation, and communication. Significant growth in enrollments were found in Spain, India, and Italy (Udemy Special Report 2020). In terms of topics enrolled in Udemy by geographic profiles, there's a massive spike of US enrollees for creative skills like Adobe Illustrator, enrollees from Spain are taking courses in piano and investing, Indian enrollees are taking business fundamentals and communication skills courses, while students from Italy are taking courses on guitar, Photoshop, and copywriting (Ederle, 2020). Most Udemy courses are paid per course and delivered via video within the platform, complete with modules lasting from 15 minutes to an hour. Coursera, another popular online course platform mostly used by students and professionals for upskilling, reported in 2022 that Philippine-based learners focused mostly on business-related courses at 62%. The online course platform also stressed that skill development strengthens island nations such as Singapore, Indonesia, and Japan. While most of Coursera's courses are paid per course, which lasts for up to 3 months, and are delivered via video within the platform, the platform also offers special subscriptions where students can pay monthly to access multiple courses.

There are numerous online copywriting courses online. Some are hosted online for free and some are sold ranging from $10 to thousands of dollars. All of these courses have varying levels of attributes. These varying attributes typically include the level of course content, the style of instructions, the overall duration of the course, the method of course delivery, and payment methods. This level of variety means that the probability of aspiring and practicing freelance copywriters enrolling in a course that suits their style and preference on the first few tries is low. Thus, it is important to analyze these attributes in order to deliver a suitable course that meets their learning requirements and preferences.

Tang conducted a conjoint analysis study in 2019 that focuses on determining the online course preferences of Chinese consumers using a 5-point Likert scale [11]. The study involved 32 kinds of online course products under the business, language, computer, engineering, courses with examinations and certificates, and interests and skills categories. The study showed that the type of the course is the most important factor and that male and female consumers have varying preferences. According to the study, male consumers focus more on the perceived ability and level of the teacher while female consumers focus more on the way courses are charged. The male consumer preferences from Tang's study are supported by Zhang et. al. [12] whose Signaling Theory approach revealed that there's an online course's rating has a positive impact on online course sales. These kinds of findings are important to a study made by Kuzmanović et al. [13], shows that student preferences are essential when it comes to designing an effective online learning system.

Despite the availability of studies on online course preferences and online course success factors [14, 15], there are no online course preference studies that focused on copywriting courses and on Filipino consumers. Considering the number of online copywriting course attributes that need to be analyzed in order to determine Filipino consumer preferences, Conjoint Analysis is deemed to be the most appropriate tool for analysis. Conjoint Analysis is a popular tool for product marketing analysis and consumer research [16] as it reveals the true preference of a consumer after placing ratings on various realistic hypothetical combinations provided in a survey [17, 18]. The results of Conjoint Analysis can help online copywriting course creators package their course in such a way that would effectively facilitate learning and mastery of copywriting skills.

On adoption and effect, a Thailand-based study on non-English major Thai students' approach to a 3-month-long synchronous online English course showed that anxiety is not a significant predictor of student performance and that there are no differences in performance between genders (Apridayani et al., 2023). While in Israel, Israeli creative arts therapies master's degree students showed a deeper appreciation and commitment to using their knowledge and skills after taking an online course on palliative and bereavement care, which was delivered in the last four sessions of the program. The courses were delivered asynchronous and synchronous, where the latter is a 3-hour session delivered by Ph.D. level experts via Zoom (Orkibi et al., 2023). An interesting study on Chinese nursing students and online courses showed that students with low family incomes and those who live in rural areas are not readily accepting of fully immersive online learning, while older male nursing students are more likely to fully immerse themselves when learning online. (Zhang et al., 2022). On the other hand, both architecture students and faculty of a university in Jordan, have shown satisfaction with online teaching when it comes to theoretical courses and dissatisfaction over the design courses, which infers that the online course's topic's level of technical involvement may influence perceived satisfaction as well (Ibrahim et al., 2021). From a commercially-driven perspective, a 2020 study by Zhang et al. shows that paid online courses are primarily purchased based on ratings and followers, while the reverse is observed on upvotes. This is tangentially affirmed by another 2020 study on free and paid e-books, which shows that consumption of e-books were primarily driven by ranking regardless of whether the e-book is paid or free, while the number of reviews doesn't affect readers' intention (Liu et al., 2020). An in-depth study of consumers' purchase intention on online paid knowledge in China has revealed that the most important purchase factor is the course creator's professionalism, followed by the course platform's information quality and interactivity, and then by the rarity of the information being sold and the course creator's perceived charisma (Zhou et al., 2022). Based on what the proponents of the studies were able to find, there are currently no studies referring to and reporting on the paid online course environment of the Philippines and paid online copywriting courses.

Considering the unprecedented reliance on online courses caused by the COVID-19 pandemic and the need to create an effective online copywriting course suited to Filipino copywriter's preferences in order to sustain livelihood and economy, there is a need to determine which online course attributes are the most preferred by Filipino copywriters in order to create a course that is appealingly packaged and pedagogically effective and subsequently streamline acquisition of knowledge and skills to generate income.

As such, the aim of this study was to determine the preferred online copywriting course attributes of Filipino copywriters of varying experience levels by uncovering the most preferred attribute, the most preferred attribute combination, the ordered attribute preferences, as well as the least preferred attribute and its combinations so that these attributes won't be adopted in future courses.

The result of this study will be highly beneficial to aspiring and practicing freelance copywriters, paid online course creators, the Philippine freelance economy, and business all over

the world who are starting to adopt digitalization to sustain its competitiveness. The insights gained from this study may serve as a foundation for improving online education by providing key course attributes that will help online courses become more appealing and more engaging, which will improve adoption and utilization especially during a pandemic.

## 2. Sustainable business model and strategies

To obtain competitive advantages, a business model may be considered for online course business sustainability. A three-tier business model adopted from Kuo [19] which was utilized in this study, is seen in Fig 2. Kuo [19] discussed how business models are highly utilized in the technology era. This is mostly used for gaining a competitive advantage, especially in an unmatched business environment. Moreover, the adaptation and innovation of the business model is a way to promote sustainability, profitability, and value in the market or industry.

From Fig 2, a three-tier business model was presented comprising competitive advantage for the first tier, innovation, resources, market, and value for the second tier, and cost, revenue, and profit for the third tier. In the essence of this study, intellectual property is a highly competitive advantage. It was stated that patent protection would be the best strategy to have a competitive advantage [20]. With that, the best strategy is to consider engagement and collaboration to promote a competitive and sustainable advantage. The way technology is being utilized in the industry would promote an innovative way of thinking among industries. Mone et al. [21] stated from their framework that innovative strategy and capabilities are the most significant factor in relation to the performance of the industry.

Following the Resource-Based Theory as discussed by Barney [22] and Kuo [19], there is a competitive advantage when it comes to intangible resources a business has such as the creation of knowledge, and then making good use of the intangible resource. This could be a great business strategy like in copywriting market. With the drastic increase in the copywriting market, especially in the 21$^{st}$ century and the COVID-19 pandemic, customer value is a significant factor in creating a competitive industry in the market [23]. Both resources and the market, therefore, lead to customer value, a relationship the copywrite businesses could take into consideration for creating strategies. As a service-center business, copywriting may consider sustainable-related content since the current generation and viewers are more inclined toward these topics [23]. Especially in the Philippine setting, even the government is an advocate of sustainability and promotes environmental conservation. This will lead to better and higher views which copywriting businesses could capitalize on.

With that, businesses should strategize considering the social, economic, and human capital in a service-related business, especially during the COVID-19 pandemic [18]. Similar to Sushil and Anbarasan [24], technology should be utilized for sustainable complexity. In application to copywriting that utilizes technology, the social value should be the main expenditure to promote the sustainable business practice. As seen in the three-tier model, to promote a sustainable business model, both monetary and nonmonetary value could be considered. In relation to the explanation of Ong et al. [18], re-strategizing, reconceptualization, proper leadership, and motivated employees would lead to a profitable and sustainable service-centered business, especially during the COVID-19 pandemic.

## 3. Methodology

This study was approved by Mapua University Research Ethics Committees. A written informed signed consent form was collected from each participant following the Data Privacy Act or Republic Act No. 10173 in the Philippines.

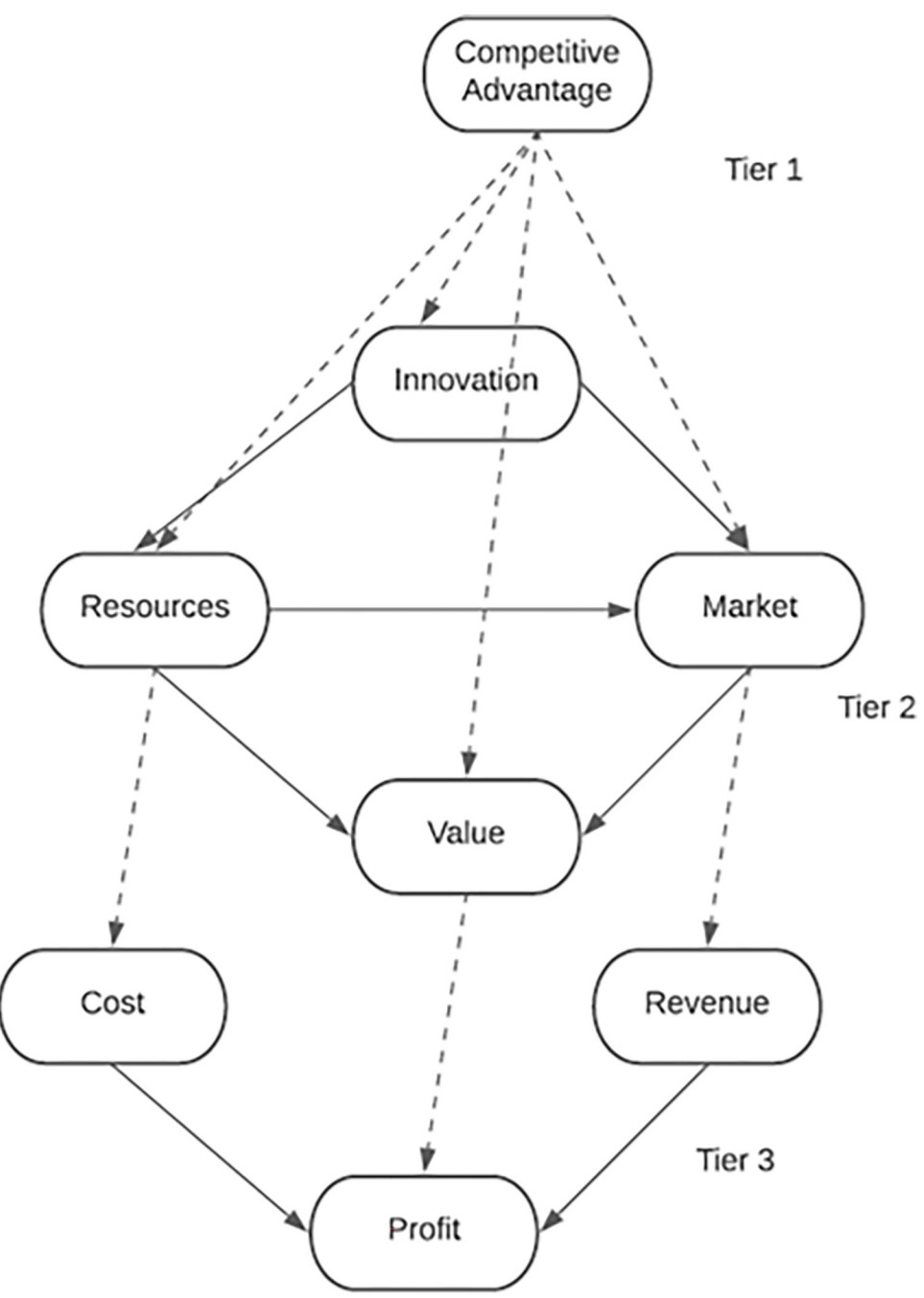

**Fig 2. Three-tier business model [19].**

Preliminary interviews were conducted with 2 Philippine-based copywriting course creators and 10 Philippine-based copywriters who took copywriting courses, which led to the determination of online course attributes and their subsequent levels. A 7-point Likert scale, repurposed as a satisfaction scale, was used in this study where 1 is labelled as 'Highly Dissatisfied' and 7 as 'Highly Satisfied'. Respondents were provided a list of online course attribute combinations, as a visual reference, prior to rating. A homogenous purposive sampling was the approach used to secure the target respondent profile. This was done by disseminating an

online survey on 'Copywriting Dojo Philippines', the first and largest copywriting community in the Philippines, therefore ensuring survey respondents are Filipino copywriters. The survey was accessible to the intended participants from July 1 to July 31, 2021. There were a total of 292 Filipino copywriting respondents who voluntarily participated in the self-administered survey.

## 3.1 Participants

Table 1 shows the demographics of the Filipino copywriters in the study. Among the 292 respondents, 67.8% were female and 32.2% were male. In the age category, the 31–40 years old group led at 36.64%, closely followed by 18–25 years old at 31.16%. The remaining age groups are 26–30 years old (17.81%), 41 to 50 years old (11.64%), below 18 years old (1.37%), and above 50 years old (1.37%). Most of the respondents reported an average monthly income of below P10,000 (35.56%). Other average monthly income listed was P10,000 to P14,999 (11.64%), P15,000 to P19,999 (13.01%), P20,000 to P29,999 (15.75%), P30,000 to P49,999 (14.38%), P50,000 to P99,999 (9.25%), and P100,000 and above (2.40%).

Profession-wise, freelancers dominate the group at 54.11%, followed by corporate employees at 22.26%. The remaining professions were spread out among entrepreneurs (5.48%), in-house copywriters (2.74%), government employees (1.71%), students (4.45%), people in sales-related activities (3.77%), and unemployed individuals (5.48%). Filipino copywriters with no direct experience working with a client dominated at 68.15%. Almost half of the respondents haven't invested in online copywriting courses (49.32%) but most of them are willing to invest P1,000—P4,999 (33.56%) to P500—P999 (31.16%).

## 3.2 Conjoint design

Table 2 presents the focal attributes of online copywriting courses in this study. This study considered course type (beginner, intermediate, advanced), payment method (per module, per course, annual membership), course style (1-on-1, video, audiobook, ebook), course module duration (< 15 minutes, < 30 minutes, 30 to 60 minutes, 1 to 3 hours), and course delivery (via email, via Facebook Group, via website). A total of 5 attributes and 17 levels were considered for this study.

The first online copywriting course attribute, the course type, refers to the level of learning content within the course. Three levels were determined, which are beginner, intermediate, and advanced. The beginner course type is focused on practical learning content that an aspiring or beginner copywriter can apply immediately. The intermediate level course type is a combination of practical content and theory, designed to help a copywriter move beyond the beginner level. The last level is advanced, which focuses on copywriting and marketing theories, giving the student a complete grasp of copywriting as a whole. These levels reflect the typical learning level segmentation of online courses. According to Zhang [25], copywriting can range from skills of beginners to advanced wherein skills on writing are being developed. Despite having lower level (beginners), advertising industries would still consider these employees as highly effective. Mahmoud [26] indicated that skills such as creativity in writing, originality, flexibility, and sensitivity to problems at hand are the levels being considered by professionals.

The second attribute, the payment method, refers to the payment options of the student in order to get access to the course. The payment method levels 'per course' and 'annual membership' reflect the typical payment options of online courses. The 'per module' level, the lightest payment option among the 3 levels, is included as an acknowledgment of the fact that 64% of Filipino families are struggling with food insecurity in 2019. Thus, payment options are highly important attributes that should be considered. The study of Danilova [27] highlighted that

**Table 1. Descriptive statistics of Filipino copywriter respondents (n = 292).**

| Characteristics | Category | N | % |
|---|---|---|---|
| Gender | Female | 198 | Female |
| | Male | 94 | Male |
| Age | Below 18 | 4 | 1.37% |
| | 18–25 | 91 | 31.16% |
| | 26–30 | 52 | 17.81% |
| | 31–40 | 107 | 36.64% |
| | 41–50 | 34 | 11.64% |
| | Above 50 | 4 | 1.37% |
| Average monthly income | Below P10,000 | 98 | 33.56% |
| | P10,000 to P14,999 | 34 | 11.64% |
| | P15,000 to P19,999 | 38 | 13.01% |
| | P20,000 to P29,999 | 46 | 15.75% |
| | P30,000 to P49,999 | 42 | 14.38% |
| | P50,000 to P99,999 | 27 | 9.25% |
| | P100,000 and above | 7 | 2.40% |
| Current profession | Freelancer | 158 | 54.11% |
| | Entrepreneur | 16 | 5.48% |
| | In-house Copywriter | 8 | 2.74% |
| | Sales-related | 11 | 3.77% |
| | Corporate Employee | 65 | 22.26% |
| | Government Employee | 5 | 1.71% |
| | Student | 13 | 4.45% |
| | Unemployed | 16 | 5.48% |
| Copywriting experience | No experience at all | 199 | 68.15% |
| | 1 client or under 3 months | 47 | 16.10% |
| | More than 1 client or more than 3 months | 46 | 15.75% |
| Total amount invested in online copywriting courses | None | 144 | 49.32% |
| | P500—P999 | 40 | 13.70% |
| | P1,000—P2,999 | 23 | 7.88% |
| | P3,000- P4,999 | 13 | 4.45% |
| | P5,000—P9,999 | 10 | 3.42% |
| | P10,000—P14,999 | 23 | 7.88% |
| | P15,000—P20,000 | 12 | 4.11% |
| | P20,000 and above | 27 | 9.25% |
| Budget for online course investment | None | 53 | 18.15% |
| | P500—P999 | 91 | 31.16% |
| | P1,000—P4,999 | 98 | 33.56% |
| | P5,000—P9,999 | 33 | 11.30% |
| | P10,000 and Above | 17 | 5.82% |

copywriters could be paid better if they possess valuable skills. Thus, it could be indicated that despite the struggle faced by citizen, attaining new skills and knowledge from online courses are still vital to gain income. Hogarth [28] also highlighted the same sentiments, and explained how skills should be developed among copywriters for them to gain more of the desired output. To which, it could be deduced that the payment made for education reciprocates their salary. People therefore are hypothesized to be sensitive with payment options for their reimbursement of investment with learning the different skills in copywriting.

**Table 2. Online course attributes.**

| Attributes | Levels |
|---|---|
| Course Type | Beginner |
| | Intermediate |
| | Advanced |
| Payment Method | Per Module |
| | Per Course |
| | Annual Membership |
| Course Style | 1 on 1 |
| | Video |
| | Audiobook |
| | Ebook |
| Module Duration | <15 minutes |
| | < 30 minutes |
| | 30 to 60 minutes |
| | 1 to 3 hours |
| Course Delivery | E-mail |
| | Facebook Group |
| | Website |

The third attribute, course style, refers to the way the course content is constructed. Four levels are presented in order to address people's varying learning styles [29]. The study suggested that certain instruction methods can be more effective for students with certain learning profiles. Similar to the study of Ong et al. [30, 31], students from different sectors have different level of course styles. It was highlighted that the effectivity of learning is also affected by the presentation and layout of courses. Thus, this was considered as one of the attributes.

The fourth attribute, course module duration, is included in order to gauge the preferred duration a student can commit to learning copywriting online, as well as their attention span's capacity. With that, <15 minutes, <30minutes, 30–60 minutes, and 1–3 hours were considered as levels. It was explained in the study of Prasetyo et al. [32] how the difference in duration of learning affects a student's learning capabilities. In addition, Ng et al. [33] highlighted the effect of the environment towards a student's learning experience. The longer the duration does not necessarily indicate better learning. This means that the preference of students towards their learning duration affects their satisfaction, motivation, and outcome [32].

Lastly, the course delivery refers to how the student wants to access and consume the course content. The 3 levels were determined based on the typical course delivery preferences. Facebook Groups are included for easy accessibility as the Philippines has 72.5 million Facebook users in 2019 or a 67% adoption rate. Website is included for course content exclusivity and access security. Finally, email is included in response to the growing trend of independent course creators sending their courses over email for convenient and organized course consumption. With the current trend in online learning, modalities of delivery have been widely considered. Dhawan [34] exasperated several modalities of online learning and presented their strengths, weaknesses, challenges, and opportunities. To which, different types of learning and modalities may be applied depending on the course type and student and teacher preference. The easiest way to deliver may seem to be effective most of the time when courses are more on self-practice like skill development.

### 3.3 Statistical analysis

A conjoint analysis with an orthogonal design was utilized to produce a reasonable number of stimuli using SPSS 25, which resulted in 29 stimuli, 93% less than the original course combinations [17, 18]. Table 3 presents the 29 stimuli evaluated by a 7-point Liker scale that represents the satisfaction level of the respondent on the presented course combination, which ranges from 1 as 'Highly dissatisfied' to 7 as 'Highly satisfied'.

The insights to be gleaned from this analysis should reveal key attributes and attribute combinations to adopt and to avoid in order to improve the appeal, adoption, and engagement of online courses, especially during a pandemic where in-person interactions are limited.

## 4. Results

Tables 4 and 5 show the respective utilities and average importance score of each online course attribute and its levels based on Filipino copywriters' responses to the survey. The average importance score showed that course style was the attribute Filipino copywriters prefer the most, which was followed by payment method, course delivery, module duration, and, lastly, course type.

**Table 3. Stimulus rank.**

| Combination | Course Type | Payment Method | Course Style | Module Duration | Course Delivery | Total | Rank |
|---|---|---|---|---|---|---|---|
| 1 | intermediate-level course | pay per course | 1-on-1 | 30 to 60 minutes per module | via Facebook Group | 0.116 | 9 |
| 2 | advanced-level course | pay per module | 1-on-1 | 30 to 60 minutes per module | via Facebook Group | -0.102 | 23 |
| 3 | beginner-level course | pay per course | audiobook | 1 to 3 Hours per module | via website | -0.018 | 17 |
| 4 | intermediate-level course | pay per module | ebook | 1 to 3 Hours per module | via Email | -0.031 | 19 |
| 5 | beginner-level course | pay per course | ebook | < 15 minutes per module | via Facebook Group | -0.036 | 21 |
| 6 | beginner-level course | pay per course | 1-on-1 | 30 to 60 minutes per module | via website | 0.174 | 6 |
| 7 | beginner-level course | pay per course | 1-on-1 | < 15 minutes per module | via Email | 0.14 | 7 |
| 8 | advanced-level course | pay per module | 1-on-1 | 1 to 3 Hours per module | via Email | 0.024 | 14 |
| 9 | beginner-level course | pay per module | video | < 15 minutes per module | via Facebook Group | 0.037 | 12 |
| 10 | intermediate-level course | pay per course | video | 1 to 3 Hours per module | via Facebook Group | 0.281 | 1 |
| 11 | beginner-level course | pay per module | 1-on-1 | < 15 minutes per module | via Email | -0.002 | 16 |
| 12 | intermediate-level course | pay per course | 1-on-1 | < 15 minutes per module | via Email | 0.202 | 4 |
| 13 | intermediate-level course | annual membership | video | < 15 minutes per module | via Email | 0.176 | 5 |
| 14 | intermediate-level course | pay per module | ebook | < 30 minutes per module | via Facebook Group | -0.122 | 24 |
| 15 | beginner-level course | pay per course | 1-on-1 | 30 to 60 minutes per module | via Email | 0.099 | 11 |
| 16 | beginner-level course | pay per module | audiobook | < 30 minutes per module | via Email | -0.281 | 29 |
| 17 | intermediate-level course | annual membership | audiobook | 30 to 60 minutes per module | via Email | -0.222 | 27 |
| 18 | intermediate-level course | pay per module | audiobook | < 15 minutes per module | via Email | -0.213 | 26 |
| 19 | intermediate-level course | pay per module | 1-on-1 | < 15 minutes per module | via website | 0.135 | 8 |
| 20 | beginner-level course | annual membership | 1-on-1 | < 30 minutes per module | via Facebook Group | -0.021 | 18 |
| 21 | intermediate-level course | pay per course | 1-on-1 | < 30 minutes per module | via website | 0.271 | 2 |
| 22 | advanced-level course | pay per course | audiobook | < 15 minutes per module | via Facebook Group | -0.192 | 25 |
| 23 | beginner-level course | pay per module | video | 30 to 60 minutes per module | via website | 0.116 | 10 |
| 24 | advanced-level course | annual membership | ebook | < 15 minutes per module | via website | -0.04 | 22 |
| 25 | beginner-level course | pay per course | ebook | 30 to 60 minutes per module | via Email | -0.032 | 20 |
| 26 | advanced-level course | pay per course | video | < 30 minutes per module | via Email | 0.204 | 3 |
| 27 | intermediate-level course | pay per module | audiobook | < 15 minutes per module | via Facebook Group | -0.258 | 28 |
| 28 | advanced-level course | pay per module | 1-on-1 | 30 to 60 minutes per module | via website | 0.018 | 15 |
| 29 | beginner-level course | annual membership | 1-on-1 | 1 to 3 Hours per module | via Facebook Group | 0.025 | 13 |

**Table 4. Online course attribute utilities.**

| Attributes | Preference | Utility Estimates | Std. Error |
|---|---|---|---|
| Course Type | Beginner | -0.016 | 0.011 |
| | Intermediate | 0.046 | 0.011 |
| | Advanced | -0.03 | 0.013 |
| Payment Method | Per Module | -0.058 | 0.011 |
| | Per Course | 0.084 | 0.011 |
| | Annual Membership | -0.026 | 0.013 |
| Course Style | 1 on 1 | 0.08 | 0.012 |
| | Video | 0.164 | 0.014 |
| | Audiobook | -0.193 | 0.014 |
| | Ebook | -0.051 | 0.014 |
| Module Duration | <15 minutes | 0.002 | 0.012 |
| | < 30 minutes | -0.004 | 0.014 |
| | 30 to 60 minutes | -0.039 | 0.014 |
| | 1 to 3 hours | 0.042 | 0.014 |
| Course Delivery | E-mail | -0.01 | 0.011 |
| | Facebook Group | -0.055 | 0.011 |
| | Website | 0.065 | 0.013 |

To identify the preferred level on each attribute, utility scores is presented in Table 4. First, the utility score for course type showed that the Intermediate level course was the most preferred level. Second, the course style attribute showed a strong preference for video, followed by 1-on-1 courses. Third, for the payment method, per course had the highest utility score. For the module duration attribute, 1 to 3 hours was the most preferred duration followed by less than 14 minutes. Lastly, the website mode of course delivery secured the highest utility score under the course delivery attribute.

Among the 29 stimuli seen in Table 3, combination 10 ranked the highest based on the total utility score. Combination 10 was an Intermediate-level video course on a Facebook group that lasts 1 to 3 hours per module and is paid per course. On the other hand, a beginner-level audiobook course that lasts less than 30 minutes per module, is delivered via email, and paid per module ranked the lowest based on the total utility score.

Table 6 shows the stimulus correlation in this study. Pearson's R-value was 0.986 and Kendall's Tau value was 0.905, suggesting a strong relationship between the observed and estimated preferences [35]. It should also be noted that 4 holdout cases were added to determine consistency among the survey responses gathered. The resulting value of 1.00 for Kendall's tau for holdouts strongly implies that the collected data were valid [35].

**Table 5. Averaged importance score.**

| Importance Values | Score |
|---|---|
| Course Type | 9.833 |
| Payment Method | 18.236 |
| Course Style | 46.007 |
| Module Duration | 10.489 |
| Course Delivery | 15.435 |

**Table 6. Stimulus correlation.**

|  | Value | Significance |
|---|---|---|
| Pearson's R | 0.986 | 0.001 |
| Kendall's Tau | 0.905 | 0.001 |
| Kendall's Tau for Holdouts | 1 | 0.021 |

## 5. Discussion

Among the 29 stimuli listed, the result of the conjoint analysis showed that what Filipino copywriters prefer from an online course is an intermediate-level video course on a Facebook group that lasts 1 to 3 hours per module and is paid per course, which had a total utility score of 0.281. On the same end, the least preferred online course combination was a beginner-level audiobook course that lasts less than 30 minutes per module, is delivered via email, and paid per module, which has a total utility score of -0.281.

Course style had the most important attribute for Filipino copywriters, based on a score of 46.007%. Under the course style attribute, the most preferred style is video followed by 1-on-1. Between the remaining course styles, e-books, and audiobooks, the latter is the least preferred. Course type was the least important attribute considered by Filipino copywriters with a score of 9.833%, the only score that fell below 10%. This is in line with the finding that videos are an important ingredient in online courses due to their ability to instruct through visual and auditory means at the same time [36]. Yoon et al. [37] expounded on the video learning behavior among students in Korea. It was seen that active learners are more inclined with video learning compared to passive ones. It was also explained by the study of Angrave et al. [38] how the learners in Illinois from online video lectures should have high engagements to have beneficial output. This indicates that for online video learning to be beneficial, students should be independent and have capabilities to handle self-paced responsibilities. This is also applicable as a result from this study in the Philippines.

The next highest attribute that Filipino copywriters consider in an online course was the payment method with a score of 18.236%. Among the 3 levels of payment method, the per-course payment method was the most preferred, followed by annual membership and per module payment methods. The third highest attribute Filipino copywriters prefer in an online course, with a score of 15.435%, was course delivery. The most preferred course delivery was via the website, followed by via email, and via a Facebook group. Video courses that are paid per course and accessed through a website are how online courses are usually sold, reflecting how Filipino copywriters prefer keeping online course consumption as it is [39]. Similar to the study of Fidalgo et al. [40] and Hussein et al. [41], students preferred the assurance of learning online with lower expenses. It was explained that saving money from expenses due to the COVID-19 pandemic was evident among students [42]. Based from the results, it could be deduced that students would want to be assured that every payment made corresponds to quality material that they can utilize to learn and develop skills.

Based on the results, the least significant among the 5 online course attributes listed were module duration and course type, with scores of 10.489% and 9.833%, respectively. Among the module durations listed, Filipino copywriters prefer modules that last 1 to 3 hours the most, followed by less than 15 minutes per module, more than 30 minutes per module, and 30 to 60 minutes per module. The result can be directly attributed to free online courses on Philippine Facebook groups typically presented as webinars, which usually last 1 to 3 hours on average, with 1 hour as the recommended webinar duration [39]. For course type, intermediate was the most preferred option, followed by beginner, then by advanced. This can be

attributed to the notion that an intermediate-level course strikes the right balance of reasonable complexity and price, which makes it an attractive choice for value-conscious Filipinos [43].

Muller and Mildenberger [44] presented that longer or shorter period of classroom hours did not have any impact of student outcome. Their study evaluated different literatures across the world during online and blended learning to decipher differences in student outputs. Subsequently, it was seen from the study of Selvaraj et al. [45] that students are active as long as the course type is delivered in platforms that can be accessed with different devices available. It could be posited from the results that copywriters would basically want to learn what is essential and needed, focus on finishing the needed lectures, and obtain required skills and knowledge for application. Since copywriting is applicable in a global perspective and industries are international, it can be deduced that the application of this finding does not encompass local settings.

As preceded discussions apply, industries are looking for copywriters who have enough skills for the job [25]. As long as copywriters could provide the required knowledge and application, industries would likely to consider them for the position [26]. Thus, the results of this study may be considered as extension and can be applied among other students or learners for copywriting worldwide. In addition, he methodology utilized, conjoint analysis can also be considered to evaluate other online learning attributes and levels, not only in copywriting, but also to other skills and knowledge needed.

## 6. Conclusions

The significant findings and contributions of this study must be considered within the context of some limitations. First, the data collection and preference measurement were made through an online survey that was deployed in a Facebook group during the COVID-19 pandemic. Considering that the Facebook group, Copywriting Dojo Philippines, is a beginner-friendly community and that the job security issues caused by the pandemic led to a surge of Facebook group members, which led to beginner-level copywriters becoming the major respondents. More experienced and advanced respondents may result in a finding that may be useful for creating strategies for retaining participants in online courses. Furthermore, the study is only focused on Filipinos who are copywriters. Future research could include data from other nationalities and other industries that will benefit from online courses as this would help provide a general course preference and become the online course standard. Moreover, clustering techniques such as K-Means may highlight also the target demographics to promote the online course. Lastly, the online course attributes that were evaluated in this study were based on online courses usually offered on Facebook groups in the Philippines. In future research, it would be more enlightening to include installment payments, course creator interactions, peer-to-peer interactions, assignments, projects, instructor reputation, and the inclusion of bonus content as part of the course attributes and levels to be considered.

It appears that Filipino copywriters are strongly opinionated on their preferences when it comes to online copywriting courses as shown by the significant gaps between utility estimates. The results of this study show that when it comes to online courses, Filipino copywriters primarily prefer watching videos, paying per course, and enjoying their course on a website. For online course creators targeting Filipino copywriters, this means they can reduce the complexity of their course-building decisions and speed up the production of their course. From a sales perspective, this also means that they can attract more students and generate more revenue with a course that matches their preferences. Competition-wise, the result of this study should lead to a healthy online course competition based on the quality of the instructor, the

curriculum, and content, which will help a lot of Filipino copywriters gain and improve their copywriting skills in the most efficient way possible. On a global scale, this also means there will be an ample supply of copywriting talent to meet the unprecedented demand caused by the COVID-19 pandemic. Therefore, these attributes and findings could be capitalized on by entrepreneurs to have higher engagement in online courses offered to heighten their businesses.

These insights are important as the restrictions that were imposed in response to the COVID-19 pandemic led to the inevitable reliance on online courses [32, 46–49], especially for aspiring Filipino copywriters who need to learn new skills to generate a new source of income and practicing Filipino copywriters who need to improve their skills to keep their jobs or take on more clients. This study utilized Conjoint Analysis with an orthogonal design to identify the online course attribute combination most preferred by Filipino copywriters. Packaging the survey through a Conjoint Analysis approach made it easier for the 292 Filipino copywriters who voluntarily participated in the online survey by limiting the selection to with a total of 29 online course combinations presented. Course style, payment method, course delivery, module duration, and course type were used as the online course attributes to be evaluated.

Overall, this study found that course style, or the course format, is significantly the most important attribute that Filipino copywriters want in an online course. It indicates that Filipino copywriters are inclined to first consider how they are going to consume the online course. The same perspective goes for the online course payment method and course delivery, which signals the need for a convenient way to access the course. Based on these findings, the proponent recommends that online course creators should prioritize their course-building process around these 3 primary attributes first in order to increase the probability of enrolment from Filipino copywriters, and more importantly, reduce the number of factors to consider when testing and optimizing their courses. With these findings, this study can be considered the first on determining the online course preferences of Filipino copywriters. The findings of this study will strongly benefit course creators, the Filipino copywriting community, and even academicians who are interested to extend the study to other nationalities and industries.

## Author Contributions

**Conceptualization:** Cheselle Jan Roldan, Yogi Tri Prasetyo, Ardvin Kester S. Ong.

**Data curation:** Ardvin Kester S. Ong.

**Formal analysis:** Cheselle Jan Roldan, Yogi Tri Prasetyo, Ardvin Kester S. Ong.

**Funding acquisition:** Yogi Tri Prasetyo.

**Investigation:** Cheselle Jan Roldan, Yogi Tri Prasetyo, Ardvin Kester S. Ong.

**Methodology:** Cheselle Jan Roldan, Yogi Tri Prasetyo, Ardvin Kester S. Ong.

**Resources:** Cheselle Jan Roldan.

**Software:** Cheselle Jan Roldan, Yogi Tri Prasetyo, Ardvin Kester S. Ong.

**Supervision:** Yogi Tri Prasetyo, Ardvin Kester S. Ong, Satria Fadil Persada, Reny Nadlifatin.

**Validation:** Irene Dyah Ayuwati, Satria Fadil Persada, Reny Nadlifatin.

**Writing – original draft:** Cheselle Jan Roldan, Yogi Tri Prasetyo, Ardvin Kester S. Ong.

**Writing – review & editing:** Irene Dyah Ayuwati, Satria Fadil Persada, Reny Nadlifatin.

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
