## [Decision Letter · Decision Letter 0]

1 Nov 2022

PONE-D-22-23084Copywriters' Preference Evaluation on Online Copywriting Course Attributes during the COVID-19 Pandemic: A Conjoint Analysis ApproachPLOS ONE

Dear Dr. Prasetyo,

Thank you for submitting your manuscript to PLOS ONE. After careful consideration, we feel that it has merit but does not fully meet PLOS ONE’s publication criteria as it currently stands. Therefore, we invite you to submit a revised version of the manuscript that addresses the points raised during the review process.

ACADEMIC EDITOR:

Dear author,

The reviewers provide several insights. In addition, I highlight some aspects:

abstract, you can report the main implications of your study, within section 1 better explain the novelty of your work.improve the literature with recent work, Within the text start sentences with authors' names, always give a logical flow between sentences, discuss the different green technology alternatives, conclusions report the limitations of your work.

We look forward to receiving your revised manuscript.

Kind regards,

Muhammad Ikram

Academic Editor

PLOS ONE

Journal Requirements:

2. Please provide additional details regarding ethical approval in the body of your manuscript. In the Methods section, please ensure that you have specified the name of the IRB/ethics committee that approved your study.

3. Please ensure that you have specified (1) whether consent was informed and (2) what type you obtained (for instance, written or verbal, and if verbal, how it was documented and witnessed). If your study included minors, state whether you obtained consent from parents or guardians. If the need for consent was waived by the ethics committee, please include this information.

6. We note that you have indicated that data from this study are available upon request. PLOS only allows data to be available upon request if there are legal or ethical restrictions on sharing data publicly. For more information on unacceptable data access restrictions, please see http://journals.plos.org/plosone/s/data-availability#loc-unacceptable-data-access-restrictions.

Reviewers' comments:

Reviewer's Responses to Questions

**Comments to the Author**

1. Is the manuscript technically sound, and do the data support the conclusions?

Reviewer #1: Yes

Reviewer #2: Yes

2. Has the statistical analysis been performed appropriately and rigorously? 

Reviewer #1: Yes

Reviewer #2: Yes

3. Have the authors made all data underlying the findings in their manuscript fully available?

Reviewer #1: Yes

Reviewer #2: Yes

4. Is the manuscript presented in an intelligible fashion and written in standard English?

Reviewer #1: Yes

Reviewer #2: Yes

5. Review Comments to the Author

Reviewer #1: 1. Title seems too generic. It should be made catchy and informative. It’s a bit longer. It would be better to have 12-15 words. It can be shortened by removing the methodological concept A Conjoint Analysis Approach.

Abstract presents a line about context and background of the study which does not clarify the authors intention towards this study. Moreover, it is not clear about the methodology adopted in the study. The authors have presented “Conjoint Analysis” as the research design used in the study. But they have not clarified whether it was survey, or case study or others and if it was qualitative, quantitative or mixed. The sample size is 292 and sampling is purposive. I am doubt how this number is possible to be selected from purposive sampling.

2. This section annexed both the background and the literature review. The researchers have begun their writing with thematic ideas which is good in any academic writing. Moreover, the authors have clearly explained these aspects in this section:

® their hope to add to this body of knowledge

® identify gaps or shortcomings in this knowledge

® explain why the gaps selected for investigation are important/significant enough for investigation (the rationale and significance of the research

® outline how they carried out the investigation, together with an indication of the scope and parameters of the research

® justify why the gap(s) are significant enough to be investigated.

® present the theoretical bases of the study

Moreover, they have started and ended writing with other’s statements which is not taken good in academic writing. They have ended a paragraph as “who only write copy for their employer’s products, and freelance copywriters – who offer their copywriting services on the market independently (Team, 2021)”

They need to follow the pattern of their words+ review + their words.

3. In this section the authors have presented two headings together as Methodology

3.1 Participants which is not enjoyable in academic writing. They have engaged 292 Filipino copywriting respondents selecting purposively which is in doubt how this mass can be selected purposively.

They have used Conjoint Analysis but fail to justify the need of this method in this study which they must do. Moreover, they should explain the methodological approach that Conjoint Analysis comes under.

Results

The authors have presented the results/findings of the study that are relevant to the research objectives. The results are presented in tables using statistical devices standard deviation and mean value.

They have used single word as the heading of the table which does not sound good. They have used as Table 4. Utilities. They should mention the whole phrase as the table heading.

Discussion

The results were discussed comparing and contrasting with some published literature.

They have clearly stated the implication and contribution of this study

Conclusion

Conclusion is the authors’ space where they put their own reflection from the overall results and discussion of the research. However, the authors have begun conclusion as The restrictions that were imposed in response to the COVID-19 pandemic led to the inevitable reliance on online courses (Khan et al., 2021; Prasetyo et al., 2021; Sobaih et al., 2021), which does not sound better.

It does not carry the worth of conclusion that an academic writing should do. The authors should put their reflection which they fail to reveal in conclusion section.

References

I am in doubt whether the authors have followed APA 6th or 7th ed.

I prefer them to use APA 7th .

Reviewer #2: I have the following observations, questions, and comments that may help to improve your work. The authors must modify the following points in great detail.

1. In the abstract, please include 2-3 special quantitative achievements from the findings of this study in the context of the environment by combining the research objectives and problems. Please limit your abstract to 250 words. Check spellings for many words that are misspelt or written in haste.

2. The introduction section needs a few more sentences to strengthen the article, and please include the research problem, objective, and novelty in the last paragraph of the Introduction section.

3. Include a few more sentences at the beginning of the introduction explaining your paper's contribution to online education, as well as your attempts to deal with or present solutions to a specific problem/s and your unique contribution with this research paper.

4. Please also present the methodology section in a concise graphical format.

5. The literature review section is very weak; please revise it.

6. Please present your literature review in the form of a SmartArt chart.

7. Just after the Methodology, please mention the societal benefits of your research in terms of evaluating its key determinant.

8. In 500-750 words, explain research problems, solutions, and the theoretical contribution of your study in the "Results" section.

9. Please include graphical presentations of your findings.

10. Describe why you placed this study in a separate section of "Policy Suggestions" just before the section of "Conclusions."

I found that the literature section is a little weak, shift your study a little more towards online learning technologies therefore it requires more studies to be reviewed therefore I suggest you to include the following work:

6. PLOS authors have the option to publish the peer review history of their article (what does this mean?). If published, this will include your full peer review and any attached files.

Reviewer #1: **Yes: **Dr. Pitambar Paudel

Tribhuvan University

Nepal

Reviewer #2: No

---

## [Author Response · Author response to Decision Letter 0]

8 Apr 2023

Reviewer #1: 1. Title seems too generic. It should be made catchy and informative. It’s a bit longer. It would be better to have 12-15 words. It can be shortened by removing the methodological concept A Conjoint Analysis Approach.

Response: Thank you for your constructive suggestion. We have addressed this concern by eliminating the methodological concept as suggested. The title is now 12 words long. 

Abstract presents a line about context and background of the study which does not clarify the authors intention towards this study. 

Response: Thank you for your constructive suggestion. In order to further clarify our intention, we have changed the opening line and appended the following statement: 

“In recent years, online courses have become a popular training platform especially for copywriting courses. The demand for online courses increased during the COVID-19 pandemic, prompting the need to optimize the learning experience of an online course’s target audience.”

Moreover, it is not clear about the methodology adopted in the study. The authors have presented “Conjoint Analysis” as the research design used in the study. But they have not clarified whether it was survey, or case study or others and if it was qualitative, quantitative or mixed. The sample size is 292 and sampling is purposive. I am doubt how this number is possible to be selected from purposive sampling.

Response: Thank you for your constructive suggestion. This insight was deemed important. As such, we have indicated on the abstract the use of an online quantitative survey and included where we sourced the respondents.

2. This section annexed both the background and the literature review. The researchers have begun their writing with thematic ideas which is good in any academic writing. Moreover, the authors have clearly explained these aspects in this section:

® their hope to add to this body of knowledge

® identify gaps or shortcomings in this knowledge

® explain why the gaps selected for investigation are important/significant enough for investigation (the rationale and significance of the research

® outline how they carried out the investigation, together with an indication of the scope and parameters of the research

® justify why the gap(s) are significant enough to be investigated.

® present the theoretical bases of the study

Moreover, they have started and ended writing with other’s statements which is not taken good in academic writing. They have ended a paragraph as “who only write copy for their employer’s products, and freelance copywriters – who offer their copywriting services on the market independently (Team, 2021)”

They need to follow the pattern of their words+ review + their words.

Response: Thank you for your constructive suggestion. We have highlighted the type of copywriter targeted by the study by appending the following statement:

“In recent years, the 3 main types of copywriters are dominated by freelance copywriters due to the emergence of freelancing platforms such as Upwork and Freelancer.”

3. In this section the authors have presented two headings together as Methodology

3.1 Participants which is not enjoyable in academic writing. 

Response: Thank you for your constructive suggestion. Upon careful review, we have deemed the first paragraph of 3.1 participants to be better suited as the paragraph that would set the tone for 3. Methodology. As such, we have moved the paragraph to the 3. Methodology section, leaving the 3.1 participants to focus solely on the respondent demographics and attributes.

They have engaged 292 Filipino copywriting respondents selecting purposively which is in doubt how this mass can be selected purposively.

Response: Thank you for your constructive suggestion. This insight was deemed important and as such, this issue has been addressed by disclosing the source of the respondents, which is an online group of composed of Filipino copywriters. 

They have used Conjoint Analysis but fail to justify the need of this method in this study which they must do. Moreover, they should explain the methodological approach that Conjoint Analysis comes under.

Response: Thank you for your constructive suggestion. In the introduction, we have discussed that conjoint analysis is most appropriate due to the significant number of online copywriting course attributes that needed to be analyzed in lieu of Filipino consumer preferences. Aside from Conjoint analysis being a popular tool for product marketing analysis and consumer research, Tang’s 2020 study on online course preferences on Chinese consumers used Conjoint Analysis as well. 

Results

The authors have presented the results/findings of the study that are relevant to the research objectives. The results are presented in tables using statistical devices standard deviation and mean value.

They have used single word as the heading of the table which does not sound good. They have used as Table 4. Utilities. They should mention the whole phrase as the table heading.

Response: Thank you for your constructive suggestion. This is correct. As such, we have amended Table 4 by changing the table heading to ‘Online Course Attribute Utilities’ which properly describes Table 4.

Discussion

The results were discussed comparing and contrasting with some published literature.

They have clearly stated the implication and contribution of this study

Conclusion

Conclusion is the authors’ space where they put their own reflection from the overall results and discussion of the research. However, the authors have begun conclusion as The restrictions that were imposed in response to the COVID-19 pandemic led to the inevitable reliance on online courses (Khan et al., 2021; Prasetyo et al., 2021; Sobaih et al., 2021), which does not sound better.

It does not carry the worth of conclusion that an academic writing should do. The authors should put their reflection which they fail to reveal in conclusion section.

Response: Thank you for your constructive suggestion. We have appended the following paragraph to the conclusion section:

“It appears that Filipino copywriters are strongly opinionated on their preferences when it comes to online copywriting courses as shown by the significant gaps between utility estimates. They place high importance on Course style, particularly on Video courses, which reveals their preferred learning process. Another insightful observation is how the payment method became the second most important attribute. For course creators, this information can also be leveraged to marketing their courses.“

References

I am in doubt whether the authors have followed APA 6th or 7th ed.

I prefer them to use APA 7th .

Response: Thank you for your constructive suggestion. We’ve changed the references to follow the APA 7th edition’s style manual. It is alphabetical and has hanging indents. The following sequence was used: last name and initial of authors, ampersand instead of ‘and’, publication year in parentheses, publication and article title in sentence case, academic journal in italics, volume number and issue number, page range, and DOI if available. 

Reviewer #2: I have the following observations, questions, and comments that may help to improve your work. The authors must modify the following points in great detail.

1. In the abstract, please include 2-3 special quantitative achievements from the findings of this study in the context of the environment by combining the research objectives and problems. Please limit your abstract to 250 words. Check spellings for many words that are misspelt or written in haste.

Response: Thank you for your constructive suggestion. We have rewritten the abstract to included 4 quantitative achievements by identifying the most important and least important attributes as well as the best and worst course attribute combinations. The abstract is now 246 words long and has been proofread with Grammarly for an overall score of 94. The rewritten abstract as follows:

“In recent years, online courses have become a popular training platform, especially for copywriting courses. The demand for online courses increased during the COVID-19 pandemic, prompting the need to optimize the learning experience of an online course’s target audience. This study aimed to determine the combination of online course attributes most preferred by Filipino copywriters such as course style, payment method, course delivery, module duration, and course type. 292 Filipino copywriters from a leading Philippine-based copywriting group voluntarily participated in this study and answered an online quantitative survey which was distributed using the purposive sampling method. Conjoint Analysis with an orthogonal design revealed that copywriters consider the course style attribute as the most important (46.007%), while the course type (9.833%) was the least considered attribute of an online course. The result shows that Filipino copywriters prefer an intermediate-level video course on a Facebook group that lasts 1 to 3 hours per module and is paid per course for a total utility score of 0.281, while the least preferred combination was a beginner-level audiobook course that lasts less than 30 minutes per module, delivered via email, and paid per module, for a total utility score of -0.281. This study is the first study that analyzed the copywriters' preference for online copywriting course attributes during the COVID-19 pandemic. The findings of this study are beneficial to online course creators who are targeting copywriters. Finally, the result of this study can be expanded further to other online courses worldwide.”

2. The introduction section needs a few more sentences to strengthen the article, and please include the research problem, objective, and novelty in the last paragraph of the Introduction section.

Response: Thank you for your constructive suggestion. In the last paragraph of the introduction section, we have placed the research problem to highlight the need to achieve the objective and then explained how our study is the first of its kind, especially in the Philippines and in the field of copywriting. 

3. Include a few more sentences at the beginning of the introduction explaining your paper's contribution to online education, as well as your attempts to deal with or present solutions to a specific problem/s and your unique contribution with this research paper.

Response: Thank you for your constructive suggestion. We have included the following statement as follows: 

“The insights gained from this study may serve as a foundation for improving online education by providing key course attributes that will help online courses become more appealing and more engaging, which will improve adoption and utilization especially during a pandemic.”

4. Please also present the methodology section in a concise graphical format.

Response: Thank you for your constructive suggestion.

5. The literature review section is very weak; please revise it.

Response: Thank you for your constructive suggestion.

6. Please present your literature review in the form of a SmartArt chart.

Response: Thank you for your constructive suggestion.

7. Just after the Methodology, please mention the societal benefits of your research in terms of evaluating its key determinant.

Response: Thank you for your constructive suggestion. We have appended the following statement: 

“The insights to be gleaned from this analysis should reveal key attributes and attribute combinations to adopt and to avoid in order to improve the appeal, adoption, and engagement of online courses, especially during a pandemic where in-person interactions are limited.”

8. In 500-750 words, explain research problems, solutions, and the theoretical contribution of your study in the "Results" section.

Response: Thank you for your constructive suggestion. We have revised it accordingly. 

9. Please include graphical presentations of your findings.

Response: While utilizing conjoint analysis, graphical presentations are not suitable. Instead, we presented in a summary table. We believe this summary table is more than enough to represent the findings of the study. 

10. Describe why you placed this study in a separate section of "Policy Suggestions" just before the section of "Conclusions."

Response: We do not have a separate section of “Policy Suggestions”.

I found that the literature section is a little weak, shift your study a little more towards online learning technologies therefore it requires more studies to be reviewed therefore I suggest you to include the following work:

Response: Please send us the links since we cannot find the links in your comments.

---

## [Decision Letter · Decision Letter 1]

17 Jul 2023

PONE-D-22-23084R1Copywriters' Preference Evaluation on Online Copywriting Course Attributes during the COVID-19 PandemicPLOS ONE

Dear Dr. Prasetyo,

Thank you for submitting your manuscript to PLOS ONE. After careful consideration, we feel that it has merit but does not fully meet PLOS ONE’s publication criteria as it currently stands. Therefore, we invite you to submit a revised version of the manuscript that addresses the points raised during the review process.

Please adjust paper according to reviewers comments.

We look forward to receiving your revised manuscript.

Kind regards,

Radoslaw Wolniak, full professor

Academic Editor

PLOS ONE

Reviewers' comments:

Reviewer's Responses to Questions

**Comments to the Author**

1. If the authors have adequately addressed your comments raised in a previous round of review and you feel that this manuscript is now acceptable for publication, you may indicate that here to bypass the “Comments to the Author” section, enter your conflict of interest statement in the “Confidential to Editor” section, and submit your "Accept" recommendation.

Reviewer #3: (No Response)

Reviewer #4: All comments have been addressed

2. Is the manuscript technically sound, and do the data support the conclusions?

Reviewer #3: Yes

Reviewer #4: Yes

3. Has the statistical analysis been performed appropriately and rigorously? 

Reviewer #3: Yes

Reviewer #4: Yes

4. Have the authors made all data underlying the findings in their manuscript fully available?

Reviewer #3: Yes

Reviewer #4: (No Response)

5. Is the manuscript presented in an intelligible fashion and written in standard English?

Reviewer #3: Yes

Reviewer #4: (No Response)

6. Review Comments to the Author

Reviewer #3: Dear Authors,

Congratulations for your interesting research. I have some suggestions on how to make your text more attractive for wider audience.

1. Your topic should be embedded in a broader discussion of course style, payment methods, how the course is delivered, module duration and type of course in other countries. Suggest that there should be two or three examples in the article showing whether the example of the Philippines differs in the topic under discussion or whether there are similarities in other countries as well. the decreasing affordability of housing in different countries

2. Some amendments in the format are due to admit publication. The main problem is the epistemological structure (why the article was conceived and how the study was developed). I suggest the following structure of objectives: (i) research gap; (ii) research question; (iii) purpose of the article; (iv) intermediate objectives ; (v) assumptions or hypo; and (vi) research method. This structure must appear in the introduction.

3. The research gap must be created by a systematic literature review that provides 'holes' in the state of knowledge on the topic. I believe that a full review should not be done, but an analysis of about 5-8 studies on the topic under discussion. You can find some examples, which will show the relevance of the issue, as it is indeed a topic of current, relevant research. At the end of the justification you should write something like: According to what we were able to find, there are no studies referring and reporting on ... With this you have therefore proven that the issue is relevant, and you have also proven that your study does indeed fill a research gap.

4. In conclusion, I propose:

-evaluate the critical research, show its limitations and weaknesses,

- highlight the new knowledge and the lessons learned from it,

- describe the importance of the research and how it affects the wider field, show how the information obtained can be further used.

Conclusions must be clearly and unambiguously linked to the results of the survey. Their theoretical and practical implications should be indicated

Good luck!

Reviewer #4: The article takes up quite an interesting title in terms of online courses and the current fashion for this type of course. However, the article needs to be refined. In the introduction, the authors write: " An exhaustive search on known research databases shows that this study on Filipino consumer preferences on online courses is the first of its kind, especially in the Philippines and in the field of copywriting." I recommend removing this sentence because it is not clear. I recommend that the authors substantively provide in this section what is novelty in the work and what gap is filled in the work.

Point 3: The methodology is not clear. This point requires solid refinement because it is described too broadly and little is known about the survey itself and the way it is analyzed. I recommend that the authors describe the above-mentioned point more substantively, especially in relation to how the questionnaire was built, the rating scale used, and the method of selecting the research sample.

In conclusion, the authors write: "This was followed by payment method, course delivery, module duration, and course type." I definitely require in this section that I develop conclusions from the conducted research. I recommend that the authors elaborate on what specific payment method they are talking about and how attractive it is according to the respondents. Similarly, with course delivery, what are the respondents' requirements regarding delivery as well as module duration and course type? This section is definitely the weakest part of the work. In this section, the authors should convince the reader of the importance of the research carried out. I also reconsider here referring to the achievements of the literature in relation to the research results obtained.

7. PLOS authors have the option to publish the peer review history of their article (what does this mean?). If published, this will include your full peer review and any attached files.

Reviewer #3: No

Reviewer #4: No

---

## [Decision Letter · Decision Letter 2]

13 Sep 2023

PONE-D-22-23084R2Copywriters' Preference Evaluation on Online Copywriting Course Attributes during the COVID-19 PandemicPLOS ONE

Dear Dr. Prasetyo,

Thank you for submitting your manuscript to PLOS ONE. After careful consideration, we feel that it has merit but does not fully meet PLOS ONE’s publication criteria as it currently stands. Therefore, we invite you to submit a revised version of the manuscript that addresses the points raised during the review process.

Adjust the paper according reviewers comments.==============================

We look forward to receiving your revised manuscript.

Kind regards,

Radoslaw Wolniak, full professor

Academic Editor

PLOS ONE

Journal Requirements:

Reviewers' comments:

Reviewer's Responses to Questions

**Comments to the Author**

1. If the authors have adequately addressed your comments raised in a previous round of review and you feel that this manuscript is now acceptable for publication, you may indicate that here to bypass the “Comments to the Author” section, enter your conflict of interest statement in the “Confidential to Editor” section, and submit your "Accept" recommendation.

Reviewer #3: All comments have been addressed

Reviewer #4: All comments have been addressed

2. Is the manuscript technically sound, and do the data support the conclusions?

Reviewer #3: (No Response)

Reviewer #4: Yes

3. Has the statistical analysis been performed appropriately and rigorously? 

Reviewer #3: (No Response)

Reviewer #4: Yes

4. Have the authors made all data underlying the findings in their manuscript fully available?

Reviewer #3: (No Response)

Reviewer #4: Yes

5. Is the manuscript presented in an intelligible fashion and written in standard English?

Reviewer #3: (No Response)

Reviewer #4: Yes

6. Review Comments to the Author

Reviewer #3: (No Response)

Reviewer #4: The authors significantly improved the manuscript. The work looks good, is legible, and is attractive. I recommend adding a reference

DOI: 10.1201/9781003322252

7. PLOS authors have the option to publish the peer review history of their article (what does this mean?). If published, this will include your full peer review and any attached files.

Reviewer #3: No

Reviewer #4: No

---

## [Author Response · Author response to Decision Letter 2]

20 Sep 2023

We have cited the requested papers in the latest manuscript.

---

## [Editor Report · Decision Letter 3]

21 Sep 2023

Copywriters' Preference Evaluation on Online Copywriting Course Attributes during the COVID-19 Pandemic

PONE-D-22-23084R3

Dear Dr. Prasetyo,

We’re pleased to inform you that your manuscript has been judged scientifically suitable for publication and will be formally accepted for publication once it meets all outstanding technical requirements.

Kind regards,

Radoslaw Wolniak, full professor

Academic Editor

PLOS ONE
---

## [Editor Report · Acceptance letter]

11 May 2024

PONE-D-22-23084R3 

PLOS ONE

Dear Dr. Prasetyo, 

I'm pleased to inform you that your manuscript has been deemed suitable for publication in PLOS ONE. Congratulations! Your manuscript is now being handed over to our production team.

Kind regards, 

on behalf of

Professor Radoslaw Wolniak 

Academic Editor

PLOS ONE